# Transcriptome and Metabolome Analysis Revealed That Exogenous Spermidine-Modulated Flavone Enhances the Heat Tolerance of Lettuce

**DOI:** 10.3390/antiox11122332

**Published:** 2022-11-25

**Authors:** Wenjing Sun, Jinghong Hao, Shuangxi Fan, Chaojie Liu, Yingyan Han

**Affiliations:** 1Beijing Key Laboratory for Agricultural Application and New Technique, College of Plant Science and Technology, Beijing University of Agriculture, Beijing 102206, China; 2Beijing Vocational College of Agriculture, Beijing 102442, China

**Keywords:** exogenous spermidine, heat stress, lettuce, metabolome, transcriptome

## Abstract

Lettuce is sensitive to high temperature, and exogenous spermidine can improve heat tolerance in lettuce, but its intrinsic mechanism is still unclear. We analyzed the effects of exogenous spermidine on the leaf physiological metabolism, transcriptome and metabolome of lettuce seedlings under high-temperature stress using the heat-sensitive lettuce variety ‘Beisansheng No. 3′ as the material. The results showed that exogenous spermidine increased the total fresh weight, total dry weight, root length, chlorophyll content and total flavonoid content, increased the activities of antioxidant enzymes such as superoxide dismutase (SOD), peroxidase (POD) and catalase (CAT), and decreased malondialdehyde (MDA) content in lettuce under high temperature stress. Transcriptome and metabolome analyses revealed 818 differentially expressed genes (DEGs) and 393 metabolites between water spray and spermidine spray treatments under high temperature stress, and 75 genes from 13 transcription factors (TF) families were included in the DEGs. The Kyoto encyclopedia of genes and genomes (KEGG) pathway enrichment analysis of DEG contains pathways for plant–pathogen interactions, photosynthesis-antennal proteins, mitogen-activated protein kinase (MAPK) signaling pathway and flavonoid biosynthesis. A total of 19 genes related to flavonoid synthesis were detected. Most of these 19 DEGs were down-regulated under high temperature stress and up-regulated after spermidine application, which may be responsible for the increase in total flavonoid content. We provide a possible source and conjecture for exploring the mechanism of exogenous spermidine-mediated heat tolerance in lettuce.

## 1. Introduction

Environmental temperatures are increasing and causing one of the most severe plant abiotic stresses. Global temperatures are projected to increase by 0.2 °C per decade, which will result in temperatures ranging from 1.8 to 4.0 °C above current levels by 2100 [1]. High-temperature stress is currently the main stress factor affecting plant growth and development. Due to the intensification of the greenhouse effect, the global temperature continues to rise, and agriculture is faced with serious challenges, which puts forward higher requirements for the heat resistance of plants [2]. High temperatures affect crop yield, shorten the life cycle of plants, accelerate senescence and cause a decrease in economic yield [2]. Heat stress leads to multiple changes in plant growth, development, physiological processes and yield, which are usually detrimental [3]. Heat stress affects the stability of various proteins, membranes, RNA species and cytoskeletal structures and alters enzymatic efficiency, impeding major physiological processes and causing metabolic imbalances in cellular responses [4,5,6,7].

In recent decades, exogenous applications of protectants, such as osmoprotectants, phytohormones, signaling molecules, and trace elements, have shown beneficial effects on plants grown at high temperatures because of their growth-promoting and antioxidant abilities [8]. Polyamines, mainly including putrescine (Put), spermidine (Spd), and spermidine (Spm), are low molecular weight natural compounds with aliphatic nitrogenous structures that are present in almost all organisms, from bacteria to plants and animals [9]. For plant growth and development, polyamines are widely involved in cell division and differentiation, root elongation, flower development, fruit ripening, leaf senescence, programmed cell death, DNA synthesis, gene transcription, protein translation, and chromatin organization [10,11,12,13,14,15,16]. Spermidine is an endogenous molecule that regulates plant growth and resists unfavorable environments [17]. Spd improves heat tolerance in tomato [18], lettuce [19], wheat [20], tall fescue [21], and rice [22]. Spd pretreatment inhibits membrane lipid peroxidation, increases antioxidant enzyme activities, superoxide dismutase (SOD), peroxidase (POD) catalase (CAT) and ascorbate peroxidase (APX), and reduces the accumulation of reactive oxygen species (ROS) and malondialdehyde (MDA) [23].

Lettuce (*Lactuca sativa* L.) is one of the most popular leafy vegetables in daily life. It originates from the Mediterranean coast and is highly intolerant to high temperatures, with 15~25 °C being the preferred growth temperature and growth being hindered above 30 °C. High temperatures are one of the main factors limiting the growth of lettuce. High temperatures always result in thinner leaves and longer internodes of lettuce, which leads to a decrease in its nutritional quality and commercial value [24].

In plants, flavonoids are important compounds that affect the color of leaves, flowers and fruits, and they exhibit antioxidant properties when plants are under stress. Flavonoids are considered to be an effective substance against various abiotic stresses because of their ability to reduce oxidative damage in organisms.

The development of transcriptomic and metabolomic technologies has increased the understanding of the network between genes, and RNA-seq has been widely used for various purposes in plant genetics research, especially transcriptome analysis, which has been used to explore a wide range of changes at the level of gene expression under various biotic and abiotic stresses and used to reveal differential gene expression in various biological processes. Additionally, metabolite analysis has become a new tool that can widely analyze components and is a powerful tool to study the changes in metabolites caused by various environmental changes [25]. Combining the joint analysis of different histologies to establish regulatory networks between genes and metabolites can provide a more comprehensive explanation of changes in species tolerance at multiple levels. The mechanism by which methyl jasmonate regulates monoterpene biosynthesis in grape berry skins was revealed by a combined transcriptome and metabolome analysis [26].

In our previous experiments, we demonstrated that exogenous spermidine could alleviate the damage to lettuce seedlings under high-temperature stress [27]; however, the molecular mechanism by which spermidine mediates the responses of lettuce to stress is not clear, so we attempted to explore the mechanistic model of its action through co-omics analysis. In this study, we further revealed the regulatory network between genes and metabolites by combining changes in the physiological properties of lettuce seedlings treated with spermidine under high-temperature stress with transcriptomic and metabolomic analyses and provided theoretical support for elucidating the mechanism of exogenous spermidine in enhancing the heat tolerance of lettuce.

## 2. Materials and Methods

### 2.1. Plant Material and Treatment

The heat-sensitive lettuce variety ‘Beisansheng No. 3′ was selected from Beijing University of Agriculture [27]. The full-grained lettuce seeds were placed in filter paper-lined Petri dishes for 48 h and then sown in hydroponic seedling trays with built-in Hoagland nutrient solution and placed in an artificial climate chamber with a photoperiod of 14/10 h, diurnal temperature of 23 °C/17 °C, relative humidity of 70~75%, and light intensity of 350 ± 10 μmol·m^−2^·s^−1^, and seedlings with uniform growth were selected. Seedlings were transplanted into 10 L hydroponic tanks with Hoagland nutrient solution, and treatments were started when the seedlings reached four leaves. Lettuce leaves were sprayed with spermidine treatment at 9:00 a.m. each day to the extent that the leaves were moist but not dripping droplets. According to our previous experiments, we set the Spd spray concentration to 1 mmol·L^−1^ [27]. The standard for spraying is to spray the leaf surface and leaf back evenly, with the sprayed liquid adhering to the leaf surface so that it is all wet but not dripping.

Experimental material handling can be seen in Table 1. Three treatments were set up in this experiment: normal temperature control (CK): temperature of 22 °C/17 °C (day/night) and spraying distilled water; high temperature stress (H): temperature of 35 °C/30 °C (day/night) and spraying distilled water; high temperature stress spraying with spermidine (HS): temperature of 35 °C/30 °C (day/night) and 1 mmol·L^−1^ Spd.

Each treatment was replicated three times. After 4 days of heat treatment of the lettuce seedlings, healthy plants were randomly selected, and the cotyledon portion of the leaves was collected. Samples were placed in liquid nitrogen at the time of collection and stored in a refrigerator at −80 °C for subsequent experiments

### 2.2. Measurement of the Physiological and Biochemical Parameters

The malondialdehyde (MDA) content was determined by the thiobarbituric acid method and calculated from the absorption values at 450, 532 and 600 nm. The chlorophyll content was determined by the method of Madhava Rao and Sresty [28]: 0.3 g of the sample was ground well with 2.5 mL of 95% ethanol, filtered and fixed in a 25 mL brown volumetric flask only.

The chloroplast pigment extract was poured into a cuvette, and 95% ethanol was used as a blank control. The optical density (OD) values were measured on a spectrophotometer (UV-5200, Shanghai, China) at 665 and 649 nm. Chlorophylls were calculated according to the following equations:Chl a content (mg/g) = [13.95OD665 − 6.88OD649] V/1000 W(1)
Chl b content (mg/g) = [24.96OD649 − 7.32OD665] V/1000 W(2)
Chl t content (mg/g) = [18.08OD649 + 6.63OD665] V/1000 W(3)

The flavonoid content was determined by the sodium nitrite-aluminum nitrate chromogenic method. The flavonoid content was determined using a kit (Solarbio, Beijing, China). After the sample was dried to a constant weight, 0.1 g was dissolved in 1 mL of the extraction solution. The sample was extracted by ultrasonication at 60 °C for 30 min and then centrifuged at 12,000 rpm for 10 min, and the supernatant was taken for measurement. The absorbance value of the sample extract at 470 nm was measured.

### 2.3. Determination of Antioxidant Enzyme Activity

Place 0.5 g of lettuce sample in a cooled mortar and add in 5 mL of phosphate buffer (pH 7.0) for grinding. The homogenate was collected and centrifuged at 4 °C for ten minutes at a force of 11,000× *g*. The supernatant was collected, and the above steps were repeated twice.

Superoxide dismutase (SOD) activity was measured by the NBT (nitrogen blue tetrazolium) photochemical reduction method [29], with 50% inhibition of NBT photochemical reduction as one enzyme activity unit (U). Peroxidase (POD) activity was determined by the guaiacol method, with an increase of 1 per minute of OD 470 nm as one enzyme activity unit (U). Catalase (CAT) activity was determined by the hydrogen peroxide UV spectrophotometric method, with an increase of 0.1 per minute OD240 nm as one enzyme activity unit (U). Ascorbate peroxidase (APX) activity was determined by measuring the rate of ascorbate oxidation at 290 nm (ε = 2.8 mM^−1^ cm^−1^) [30].

### 2.4. Metabolite Measurement and Quantification

The tissue sample (25 mg) was placed in a 1.5 mL tube with 800 µL of precooled precipitant (methanol: acetonitrile: distilled water = 2:2:1) and two small steel beads, then it was ground in a grinder (60 Hz, 4 min). After the removal of the beads, it was sonicated in an ice bath for 10 min (power 80 HZ), allowed to rest for 120 min at −20 °C, and then it was centrifuged for 15 min (25,000× *g*, 4 °C) and 600 µL of the supernatant was collected, and it was repeated once. The supernatant was placed in a freeze extractor to drain. Then, 600 µL of 10% methanol solution was added, it was placed in an ice bath and sonicated for 10 min (power 80 HZ) and then centrifuged for 15 min (25,000× *g*, 4 °C). The supernatant was collected, and 50 µL of each sample was injected into a QC using an ACQUITYUPLCHSST3 column (100 mm × 2.1 mm, 1.8 µm, Waters, Wilmslow, UK) for chromatographic separation, and the small molecules that eluted from the column were collected in positive and negative ion modes using high-resolution tandem mass spectrometry (Xevo G2-XS QTOF Waters, Wilmslow, UK). Peak extraction was mainly implemented by the commercial software Progenesis QI (version 2.2. Newcastle, UK), including peak alignment, peak extraction, normalization, deconvolution and compound identification steps. Three biological replicates were performed for each sample.

### 2.5. RNA Extraction and Quality Testing

The steps of total RNA extraction, RNA purity assessment, library construction, library quality control, and up sequencing were performed at UW (https://www.genomics.cn, accessed on 1 August 2019), following their standard procedures.

### 2.6. RNA-Seq Data Analysis

The high-throughput sequencing offline data (Raw Data) were filtered to obtain high-quality data (Clean Data), and the data were filtered using UW’s self-developed filtering software SOAPnuke (v1.4.0. Shenzhen, China) for statistics and trimmomatic (v0.36. Dortmund, Germany) for filtering. The clean reads were compared to the reference genome sequence using HISAT (Hierarchical Indexing for Spliced Alignment of Transcripts, http://www.ccb.jhu.edu/software/hisat, accessed on 1 August 2019). The clean reads were aligned to the genomic sequences using Bowtie2, and then the gene expression levels were calculated for each sample using RSEM (http://deweylab.biostat.wisc.edu/rsem/rsem-calculate-expression.html, accessed on 1 August 2019). FPKM (Fragments Per Kilobases per Million reads) was used to express the gene expression levels. We defined genes with more than a two-fold difference and Q-value ≤ 0.001 to be screened as significantly differentially expressed genes (fold change ≥ 2 and adjusted *p* value ≤ 0.001). Based on the annotation results of GO (http://geneontology.org/, accessed on 1 August 2019) and KEGG (http://www.genome.jp/kegg, accessed on 1 August 2019) and the official classification, we functionally classified the differentially expressed genes and performed enrichment analysis using the phyper function in R software, with FDR correction for *p* values. Functions with Q values ≤ 0.05 were considered significantly enriched.

### 2.7. Quantitative Real-Time PCR

Total leaf RNA extraction was performed using a kit (Yueyang Hua, Yueyang, China, Cat:0416-50gk). The first cDNA strand was synthesized using a reverse transcription kit (Tiangen, Beijing, China. Cat:KR118-02). The primer sequences of the related genes were downloaded from the GenBank library of NCBI. The primer design is shown in Appendix A. qRT-PCR analysis was performed using the reverse transcription product cDNA as the template and 18S as the internal reference gene. qRT-PCR was performed using the TB Green Premix Ex Taq II (2×) (Tli RNaseH Plus) kit (TaKaRa, Kyoto, Japan) in the CFX96 Real-Time PCR Detection System instrument (Bio-Rad Laboratories, Hercules, CA, USA). The reaction system was 20 μL, consisting of 9 μL TB Green Premix Ex Taq II (TliRNaseH Plus), 7 μL ddH_2_O, 2 μL cDNA template and 1 μL forward and reverse primers. The cycling progression was as follows: 95 °C for 3 min; 95 °C for 10 s and 56 °C for 30 s, for a total of 39 cycles. Gene expression change ploidy analysis was calculated using 2^−ΔΔCt^, and relative mRNA expression levels were normalized using 18S. The qRT-PCR validation of the DEGs is shown in Appendix A.

### 2.8. Statistical Analysis

One-way ANOVA was performed on all data using SPSS 22.0 software (SPSS Inc., Chicago, IL, USA), and Duncan’s test was used to test the significance of the differences between samples (*p* < 0.05).

## 3. Results

### 3.1. Effect of Exogenous Spermidine on the Growth and MDA and Chlorophyll Contents of Lettuce under High-Temperature Stress

As shown in Table 2, high-temperature stress significantly reduced the total fresh weight, shoot fresh weight, root length, total dry weight, and root-to-shoot ratio of lettuce, while the application of spermidine alleviated the damage to the total fresh weight, root length, and total dry weight under high-temperature stress, but had no significant effect on shoot fresh weight, root fresh weight, plant height and root-to-shoot ratio.

From Figure 1A, it can be seen that the leaves of lettuce seedlings under high-temperature stress were elongated, appeared to be twitching, root growth was weak, fewer roots were produced, the main roots were short, biomass accumulation was reduced, which indicated that high temperature stress inhibited the growth condition and organic matter accumulation of lettuce seedlings. At the same time, the MDA content in the leaves increased under high temperature, and the increased MDA content reflected, to some extent, the increase in membrane lipid peroxidation. In the leaves of seedlings sprayed with spermidine, these effects were ameliorated to some extent. In addition, exogenous spermidine also affected the photosynthetic pigment content of the lettuce leaves. Although the total chlorophyll content of the leaves increased under high-temperature stress, the contents of chlorophyll a and chlorophyll b did not change significantly, while the application of spermidine significantly increased the contents of chlorophyll a and total chlorophyll (Figure 1C–E).

In conclusion, exogenous spermidine increased the total fresh weight, total dry weight, and root length, reduced the MDA content, and enhanced the chlorophyllscontent of the lettuce under high-temperature stress, alleviating the damage of high temperature on growth and physiological indexes of lettuce, providing a preliminary basis for us to further explore the possible mechanism of spermidine-mediated enhancement of heat tolerance of lettuce.

### 3.2. Effect of Spermidine on the Antioxidant Enzyme Activity of Lettuce under High-Temperature Stress

Under high temperatures, SOD activity decreased in control plants sprayed with distilled water; however, SOD activity increased in plants treated with spermidine (Figure 2A). Similar to SOD activity, spermidine treatment increased CAT activity in lettuce under high temperature compared with deionized water spray (Figure 2C); compared with the control, high temperature stress had no significant effect on POD activity but spraying with spermidine under high temperature stress still significantly increased its activity compared with deionized water spray (Figure 2B). However, for APX, there was no significant effect of high temperature stress compared to the ambient control, and neither deionized water nor spermine spraying had any significant effect on its activity under high temperature stress (Figure 2D). This suggests that exogenous spermidine can withstand high temperature stress by regulating the activities of antioxidant enzymes such as SOD, CAT and POD, and may therefore attenuate oxidative damage in lettuce leaf cells.

### 3.3. Transcriptome Data Quality Analysis

We constructed nine sequencing libraries with three replicates of each treatment. The number of raw reads per sample ranged from 45.57 to 47.33 million, as shown in Appendix A. Clean reads exceeded 42.33 million for all samples except for some low-quality reads, splices and fuzzy nucleotides. The number of clear reads obtained from the cDNA libraries of each experiment indicates that the gene abundance and transcript length are sufficient. In our libraries, the percentage of clean reads was not less than 90%, with Q20 and Q30 values exceeding 96.66% and 88.09% for all samples, respectively. This indicates that the data quality of the transcriptome is sufficient (Appendix A).

### 3.4. Metabolomics Assay

The type and content of metabolites change under different stimuli leading to phenotypic changes. To investigate the effect of exogenous spermidine on changes in the lettuce metabolome under heat stress, we analyzed the major and minor metabolites in leaves. We quantified the relative levels of all filtered ions and performed PCA analysis (Figure 3B). As shown in the figure, the first two principal components explained 48% of the total variation, indicating a clear separation between water and spermine treatments under heat stress, suggesting that the identified metabolites play an important role in mitigating heat tolerance in lettuce under the influence of exogenous spermine(Figure 3B).Then, PLS-DA models (H vs. CK, HS vs. CK, HS vs. H) between each two groups were established using metaX software to screen out different metabolites, and when the values of model parameters (R2 and Q2) were high, it indicated that the current PLS-DA model was more reliable(Table 3). To further identify potential metabolites for water treatment and spermine treatment under high temperature stress, “candidate” metabolites with the following characteristics were included in the analysis with the screening conditions: (1) VIP ≥ 1; (2) fold change ≥ 1.2 or ≤ 0.8333; (3) *p*-value < 0.05, and the three were taken to intersect to obtain the common metabolite as the differential metabolite.

Compared with CK, 897 metabolites were identified under high temperature treatment and could be classified into 14 classes, while 1052 differential metabolites were identified in HSvsCK and classified into 15 classes; we obtained a total of 307 differential metabolites in HSvsH Appendix A). These metabolites included 7 alkaloids and derivatives, 75 benzenoids, 3 hydrocarbon derivatives, 60 lipids and lipid-like molecules, 8 nucleosides, nucleotides and analogues, 56 organic acids and derivatives, 7 organic nitrogen compounds, 26 organic oxygen compounds, 49 organic heterocyclic compounds and 16 phenylpropanoids and polyketides (Figure 3A).

In the comparison of the different treatments, “organic acids and their derivatives”, “organic heterocyclic compounds”, “lipids and lipid-like molecules” and “benzenes” were noted. Similarly, we found a number of “phenyl propane and polyketide compounds” in HS vs. H, and the phenyl propane pathway is the main pathway for the synthesis of flavonoids. It is noteworthy that we also found KEGG enrichment in the transcriptome for the flavonoid biosynthesis pathways, and the relative expression of most compounds was higher in the high-temperature applied spermidine treatment than in the high-temperature sprayed water treatment, as shown by clustering analysis and volcano plots (Figure 3C,D). This may further suggest that exogenous spermidine enhances the heat tolerance of lettuce by affecting flavonoid biosynthesis.

### 3.5. Analysis of DEGs

The number of DEG groups among the three treatments are shown, and overlapping parts indicate the intersection of different combinations (Figure 4B). There were 2101 genes upregulated and 960 genes downregulated in lettuce leaves under high-temperature stress compared to the control (Figure 4C). Comparing the treatment with spermidine spray to distilled water, 818 genes were upregulated, and 284 genes were downregulated. In addition, a total of 2489 genes were upregulated and 717 genes were downregulated in the high-temperature treatment with spermidine compared to the control. However, when comparing among the groups the overlapping relationships of the differentially expressed genes, there were 3061 genes, 1102 genes and 2306 DEGs expressed in the comparison of HvCK, HS vs. CK and HS vs. H, respectively, while 151 genes were expressed in common among the three treatment groups. These results suggest that spermidine affects the response of plant genes to high-temperature stress to some extent, leading to changes in the number and type of DEGs.

### 3.6. GO Enrichment Analysis

To further understand the DEG functions of H and HS, GO functional annotation of DEGs was performed (Figure 5). A Q value (padj) < 0.05 was set as the threshold for significant enrichment. According to the GO database, genes can be classified according to biological process (BP), cytological component (CC) and molecular function (MF) categories. GO functional enrichment reveals the enrichment of distinct functional entries in differentially expressed genes, annotating differentially expressed genes to individual biological functions, so that we can understand the linkage between analyzing different differentially expressed genes and biological functions.

In this analysis, we significantly enriched the subcategories of differentially expressed genes in the GO database. In the DEGs of CK compared with H, 28.48%, 38.77% and 32.75% were categorized as biological processes, cellular components and molecular functions, respectively (Figure 5A). However, 30.32%, 37.29% and 32.39% of the DEGs in the HS versus H comparison were categorized as biological processes, cellular components and molecular functions, respectively (Figure 5B), these values were observed in 28.87%, 39.04% and 32.09% of the DEGs in the CK versus HS comparison (Figure 5C).

### 3.7. KEGG Enrichment Analysis

To understand the metabolic or signaling pathways of spermidine involved in high-temperature stress, all differentially expressed genes were compared with the Kyoto encyclopedia of genes and genomes (KEGG) database to get the matched KEGG Orthology (KO). The KEGG enrichment analysis of different genes in each treatment group is shown in Figure 6. The vertical coordinates in the figure represent the KEGG pathway, and the horizontal coordinates are the enrichment factors. The larger the enrichment factor, the greater the degree of enrichment; the larger the point, the greater the number of differentially expressed genes enriched in the pathway; the bluer the point, the more significant the enrichment. The KEGG enrichment analysis of DEGs in the H and CK treatments is shown in Figure 6A. The DEGs between the H and CK treatments mainly focused on the biosynthesis of sesquiterpenes and triterpenes; biosynthesis of keratin, sulfites and waxes; fatty acid metabolism; biosynthesis of unsaturated fatty acids; starch and sucrose metabolism; degradation of other polysaccharides; phytohormone signaling; fatty acid degradation and biosynthesis; and biotin metabolism.

The KEGG analysis of DEGs between the HS and CK enrichment analyses is shown in Figure 6C. The DEGs between HS and CK were mainly involved in plant-pathogen interactions, sesquiterpene and triterpene biosynthesis, the MAPK signaling pathway, keratin, sulfite and wax biosynthesis, other polysaccharide degradation, phytohormone signaling, and unsaturated fatty acid biosynthesis.

The KEGG enrichment analysis of DEGs between HS and H is shown in Figure 6B. As shown in the figure, the DEGs between HS and H were mainly focused on unsaturated fatty acid biosynthesis, plant-pathogen interactions, sesquiterpene and triterpene biosynthesis, fatty acid metabolism, photosynthesis-antennal proteins, the MAPK signaling pathway, linolenic acid metabolism, and flavonoid biosynthesis. These results suggest that exogenous spermidine may protect against high temperature stress by improving photosynthesis, affecting and participating in signal transduction, and regulating flavonoid biosynthesis in lettuce.

### 3.8. Transcription Factors

The different transcription factors of differentially expressed genes (DEGs) were different in the different treatment comparisons (Figure 7). Seventeen transcription factor families, including 160 TF genes, were identified in the high-temperature treatment compared with control-treated lettuce seedlings, and the family with the highest number of transcription factors among the differentially expressed genes (DEGs) was AP2-EREBP, followed by the MYB, NAC, MADS, WRKY, SBP, and bHLH families (Figure 7A). In addition, a total of 172 TF genes from 28 transcription factor families, mainly including 37 AP2-EREBPs, 19 MYBs, 16 WRKYs, 14 NACs, and 12 bHLH, were found in the comparison of high-temperature spermidine treatment with the control treatment, while 75 TFs from 13 TF families were found in the comparison of the high-temperature spermidine treatment with the high-temperature control treatment. The family with the highest number of transcription factors among the differentially expressed genes (DEGs) was AP2-EREBP, followed by the WRKY, MYB, and NAM families (Figure 7B,C). Our experimental results revealed that most of the transcription factors were upregulated after spraying with spermidine under high-temperature stress (Figure 7D), suggesting that it may alleviate the damage caused by high-temperature stress mainly by upregulating the expression of AP2, MYB, WRKY and other transcription factors and thus further regulate the expression of the corresponding genes in lettuce.

### 3.9. Spermidine Regulates the Metabolism of Flavonoids under High-Temperature Stress

As shown in Figure 8A, spraying spermidine increased the total flavonoid content in leaves under high-temperature stress. A total of 19 DEGs rich in flavonoid-related metabolic pathways were identified in HS vs. H, encoding six enzymes related to flavonoid synthesis, including hydroxycinnamoyltransferase (HCT), flavonol synthase (FLS), caffeoyl coenzyme A methyltransferase (CCoAOMT) dihydroflavonol reductase (DFR), chalcone synthase (CHS), and colorless anthocyanin dioxygenase (LDOX), which are involved in the regulation of the synthesis of flavonoid substances such as dihydroflavonol, flavonol, colorless anthocyanin, and chalcone (Figure 8B). Among them, 12 genes significantly increased their expression under high-temperature stress after the application of spermidine, and half of these 12 genes enhanced their expression by more than 2-fold, which may be responsible for the increase in total flavonoid content. These results, in agreement with our previous results, suggest that exogenous spermidine regulates the synthesis of flavonoid substances in lettuce leaves, thereby affecting their tolerance to high-temperature stress.

## 4. Discussion

High temperature (HT) is a widespread environmental stress that affects most plants at all periods of growth and can limit plant growth and development and reduce productivity. Plant growth and development involve many temperature-sensitive biochemical responses [31]. One approach to dealing with the adverse effects of heat stress may involve exploring some molecules that have the potential to protect plants from the deleterious effects of HT. Polyamines (PAs) are low molecular weight aliphatic amines and organic polycations found in a variety of organisms from bacteria to plants and animals [32]. They also play an important role in plant responses to abiotic stresses. Our previous studies have shown that high-temperature stress limits normal growth and leads to the accumulation of unwanted ROS, altering the enzymatic and nonenzymatic antioxidant activity of lettuce seedlings [27], while exogenous spermidine restores growth and photosynthesis by improving carbon metabolism, thereby increasing stress tolerance [33].

In our experiments, heat stress resulted in wilting of lettuce, curling of leaves, a reduction in biomass, and an increase in malondialdehyde content. The application of spermidine improved the growth of lettuce leaves, reduced the malondialdehyde content under heat stress, and increased the chlorophyll content. Previous studies showed that Spd treatment significantly promoted FW and DW in white clover seedlings under PEG-simulated water stress conditions [34]. Zeng found a significant decrease in aboveground FW and DW of hybrid rice under low-temperature stress, while Spd-treated plants showed a significant increase in these indices by 30.48% and 18.19% under low temperature conditions [35]. Similar to these reports, our results confirm the role of spermidine in the protection of lettuce biomass.

Chlorophyll is an important indicator of the ability of leaves to maintain their green color, and the decrease in chlorophyll concentration is generally considered to be a response mechanism to reduced light uptake by chloroplasts under stress; furthermore, the decrease in chlorophyll content during stress may be related to impaired chlorophyll synthesis or pigment protein degradation and could be a result of ROS production [36]. The results in Figure 2 and Figure 3 indicate that exogenous spermidine promoted the enhancement of antioxidant enzyme activity with increased chlorophyllin lettuce seedlings, which may be one of the reasons for their increased tolerance of heat stress. Our previous study also confirmed that spraying exogenous spermine at high temperatures increased chlorophyll a and b content in lettuce leaves and was able to avoid oxidative damage to chloroplasts [37]. These results suggest that Spd spraying can maintain chlorophyll and content to support light energy capture and transport in lettuce under high temperature stress. Higher photosynthesis rates promote flavonoid accumulation because flavonoids are synthesized in chloroplasts and flavonoids are hypothesized to be positively correlated with photosynthesis [38,39,40]. Therefore, we hypothesized that enhanced photosynthetic activity in lettuce leaves increased the production of primary and secondary metabolites, including flavonoids.

Crop tolerance to HT stress is associated with an increase in antioxidant capacity [41]. Previous studies have shown that Spm pretreatment increased antioxidant enzyme activities under HT and drought stress conditions, while the activity levels of CAT, POD and SOD were consistently higher in Spm pretreated seedlings than in the control [42]. In our experiments, high-temperature stress affected POD and SOD activities in lettuce leaves, while the application of spermidine increased CAT, POD and SOD activities without a significant effect on APX. This may be due to the fact that antioxidant enzymes differ in their perception of the degree and duration of temperature, and thus different antioxidant enzymes vary over different temperature ranges.

Upregulation of many genes has been reported to help plants resist stress conditions that lead to plant adaptation [43]. Under various biotic and abiotic stresses, plants are able to receive external signals and respond to the stress by associating various internal pathways to transmit information to downstream molecules through their respective methods. The main molecular mechanisms underlying the response to salt stress were previously revealed by transcriptome sequencing analysis [44]; physiological and transcriptomic approaches were used in grapes to explore the effect of exogenous monocrotaline lactones on drought stress at the transcriptional level, among others [45]. A combined transcriptomic and metabolomic analysis also provided an enhanced understanding of the regulatory network between certain specific genes and compounds in broccoli species under selenate treatment [46]. Therefore, in our study, we attempted to investigate the response of spermidine to differential gene expression patterns, expression of different transcription factors, and plant secondary metabolites and related genes in lettuce seedlings under high temperature stress by studying RNA-sequence analysis and metabolite components.

Signaling molecules are involved in the activation of many stress-responsive genes, and various signal transduction molecules associated with the activation of stress-responsive genes exist depending on the plant type and type of stress. Among them, widely used are Ca-dependent protein kinases (CDPKs), mitogen-activated protein kinases (MPKs), NO, sugar (as signaling molecules), and phytohormones [47]. These molecules, along with transcription factors, activate stress response genes. Once activated, stress response genes reactivate essential enzymes and structural proteins, which contribute to the detoxification of ROS (by activating detoxifying enzymes and free radical scavengers) to maintain cellular homeostasis [48]. The available data suggest that some signaling molecules may lead to an increase in cellular antioxidant capacity [49,50].

TFs stimulate and regulate multiple stress response pathways in plants subjected to high-temperature stress [48,49]. Flavonoids are plant polyphenol secondary metabolites with a wide range of physiological functions [51]. The flavonoid pathway is derived from the general phenylpropane pathway, and flavonoid biosynthetic genes are regulated by interactions between different TF families [44,46]. For example, genes involved in the anthocyanin and condensed tannin pathways are regulated by R2R3MYB, bHLH, and WD40 proteins (MYB-bHLH-WD40, MBW complex) [52]. Flavonoid synthesis genes, such as F3H, F3′H and FLS, are regulated by MYB genes [53]. In the present study, AP2-EREBP, WRKY, MYB, and NAM were the most abundantly expressed transcription factor families after spermidine treatment (Figure 6), and in agreement with previous studies, spermidine may have regulated the MYB family and thus the expression of genes related to the flavonoid metabolic pathway and, in this way, coordinated the total flavonoid content. However, the regulatory functions of these TFs in flavonoid biosynthesis need to be further investigated.

Flavonoids may protect plants from oxidative stress by blocking ROS production through their ability to chelate metal ions and scavenge ROS, thus achieving their antioxidant function [54,55,56]. Moreover, since the synthesis of flavonoids is performed in chloroplasts, the increased chlorophyll content and antioxidant enzyme activity in our experimental results suggest that spermidine may be able to maintain high light energy capture and transport by increasing chlorophyll content and avoiding oxidative loss of chloroplasts to ensure smooth synthesis of flavonoids, which in turn may protect organelles from oxidative damage [39,40]. It has been shown that excessive accumulation of flavonoids with higher free radical scavenging activity in transgenic Arabidopsis enhances tolerance to drought stress [57]. In our study, KEGG enrichment analysis of DEGs showed significant enrichment in the “flavonoid biosynthesis” pathway (Figure 6B) and six enzymes related to the flavonoid synthesis pathway (oxalate hydroxycinnamyltransferase (HCT), flavonol synthase (FLS), CCoAMOT (caffeoyl-coenzyme A methyltransferase), chalcocyanine, and chalcocyanine), chalcone synthase (CHS), and colorless anthocyanin dioxygenase (LDOX)) were differentially expressed. Similarly, most of the representative genes involved in these pathways were downregulated under high-temperature stress and upregulated under high-temperature spermidine treatment (Figure 8 and Appendix A), which may be responsible for the elevated total flavonoid concentration (Figure 8A). In other words, the elevated flavonoid concentration may have mitigated the effect of high-temperature stress on lettuce leaves to some extent.

## 5. Conclusions

Exogenous spermidine increased the content of chlorophylls and antioxidant enzymes and decreased the content of malondialdehyde under high-temperature stress. By integrating transcriptome and metabolite analysis, we found that spermidine may be involved in or induce the expression of TF families such as AP2-EREBP, WRKY, MYB, and NAM to transmit information, and combined with the metabolic pathways identified in KEGG enrichment, spermidine influenced the biosynthesis of flavonoid synthesis and ultimately alleviated the damage to lettuce from high-temperature stress. A hypothetical model of exogenous spermidine-mediated heat stress tolerance in lettuce is shown in Figure 9, but its specific regulatory mechanism needs further exploration and experimental corroboration. Exogenous spermidine may be a promising approach and may enhance the heat tolerance of lettuce by regulating various factors such as growth, physiology, molecular activities and metabolite accumulation under high temperature conditions.

## Figures and Tables

**Figure 1 antioxidants-11-02332-f001:**
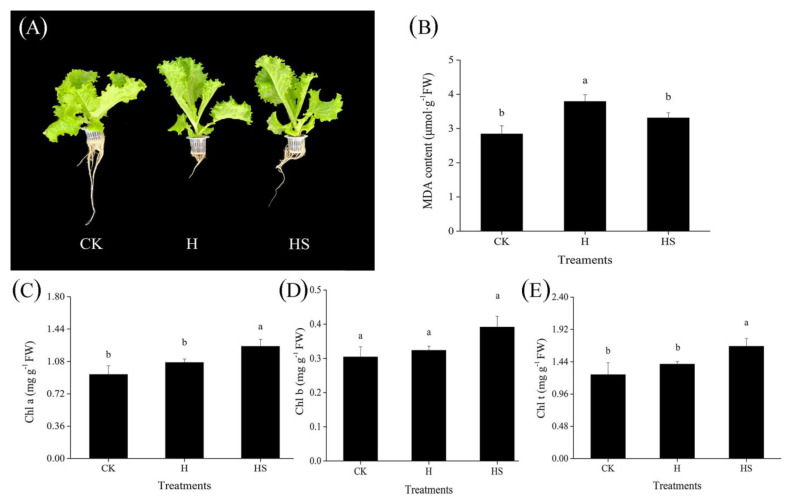
Effects of exogenous spermidine on the morphology, MDA content and chlorophyll content of lettuce under high temperature. (**A**) Phenotypic map. (**B**) MDA content. (**C**) Chlorophyll a content. (**D**) Chlorophyll b content. (**E**) Total chlorophyll content. Values above each vertical bar followed by different letters show significant differences (*p* < 0.05). The highest value was labeled as a, and those with significant differences were labeled as b in that order. CK: 22 °C/17 °C, distilled water; H: 35 °C/30 °C, distilled water; HS: 35 °C/30 °C, distilled 1 mM Spd.

**Figure 2 antioxidants-11-02332-f002:**
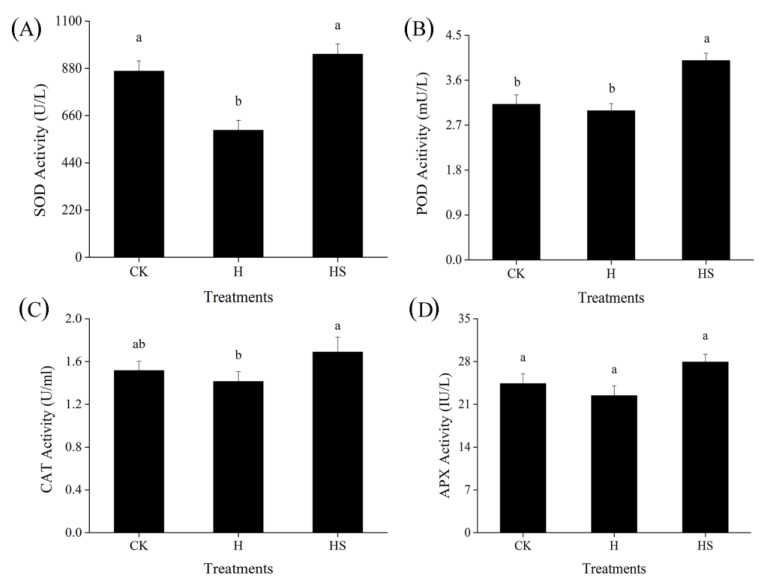
Effects of exogenous spermidine on the activities of antioxidant enzymes in lettuce under high-temperature stress. (**A**) SOD activity, (**B**) POD activity, (**C**) CAT activity, and (**D**) APX activity. Values above each vertical bar followed by different letters show significant differences (*p* < 0.05). The highest value was labeled as a, and those with significant differences were labeled as b in that order. CK: 22 °C/17 °C, distilled water; H: 35 °C/30 °C, distilled water; HS: 35 °C/30 °C, distilled 1 mM Spd.

**Figure 3 antioxidants-11-02332-f003:**
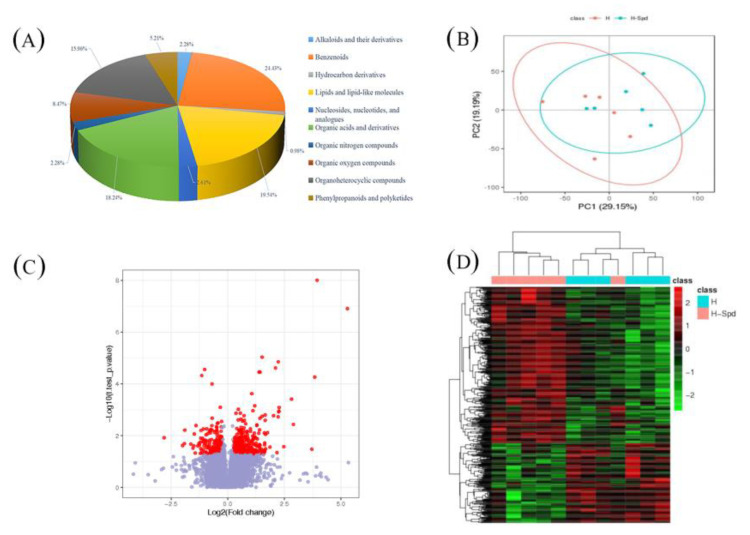
Differential metabolome analysis in HSvsH. (**A**) Differential metabolite composition. (**B**) PCA model for metabolic data. (**C**) Volcano plot of differential metabolites. (**D**) Heat map for cluster analysis of differential metabolites. CK: 22 °C/17 °C, distilled water; H: 35 °C/30 °C, distilled water; HS: 35 °C/30 °C, distilled 1 mM Spd.

**Figure 4 antioxidants-11-02332-f004:**
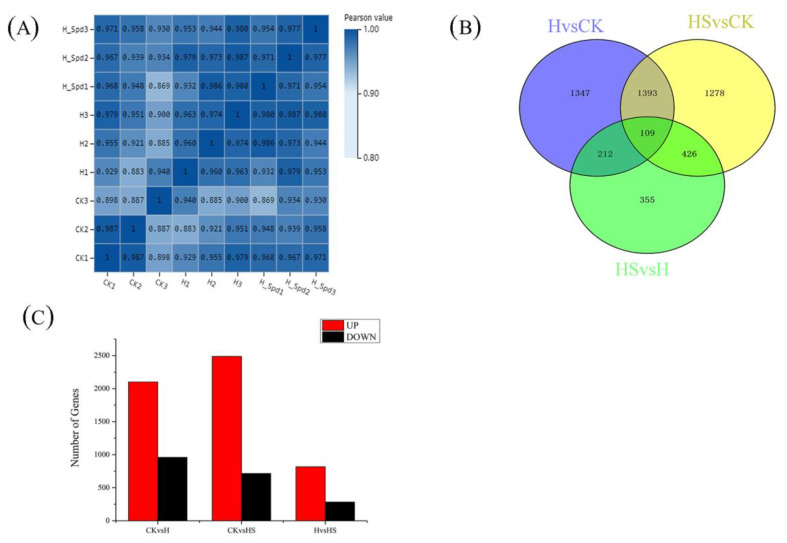
Analysis of DEGs in lettuce under high-temperature stress with exogenous spermidine. (**A**) Transcriptome sample correlation map. (**B**) Venn diagrams of DEGs. (**C**) Numbers of DEGs. CK: 22 °C/17 °C, distilled water; H: 35 °C/30 °C, distilled water; HS: 35 °C/30 °C, distilled 1 mM Spd.

**Figure 5 antioxidants-11-02332-f005:**
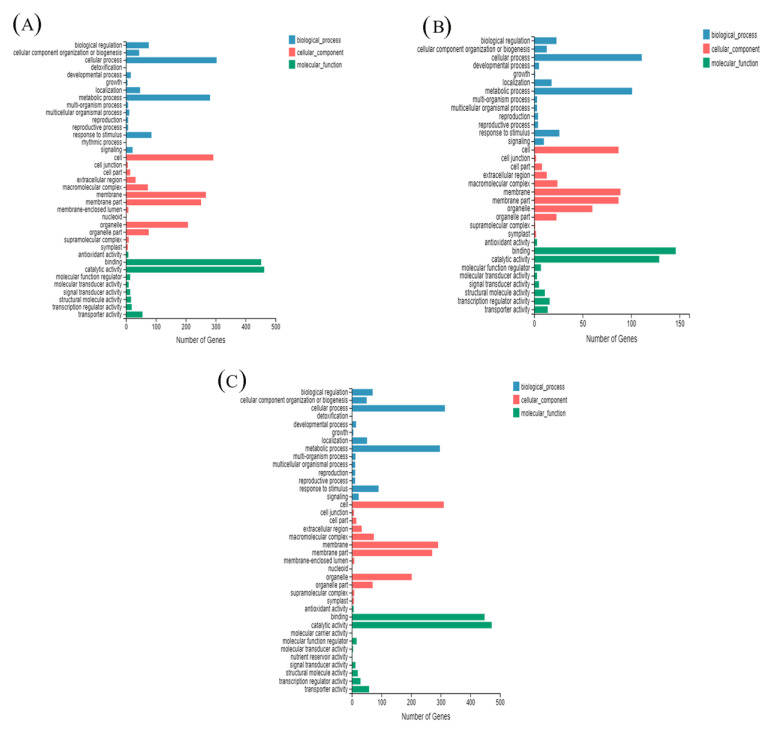
Gene Ontology (GO) classification and distribution of GO annotated genes. (**A**) H vs. CK; (**B**) HS vs. H; (**C**) HS vs. CK. CK: 22 °C/17 °C, distilled water; H: 35 °C/30 °C, distilled water; HS: 35 °C/30 °C, distilled 1 mM Spd.

**Figure 6 antioxidants-11-02332-f006:**
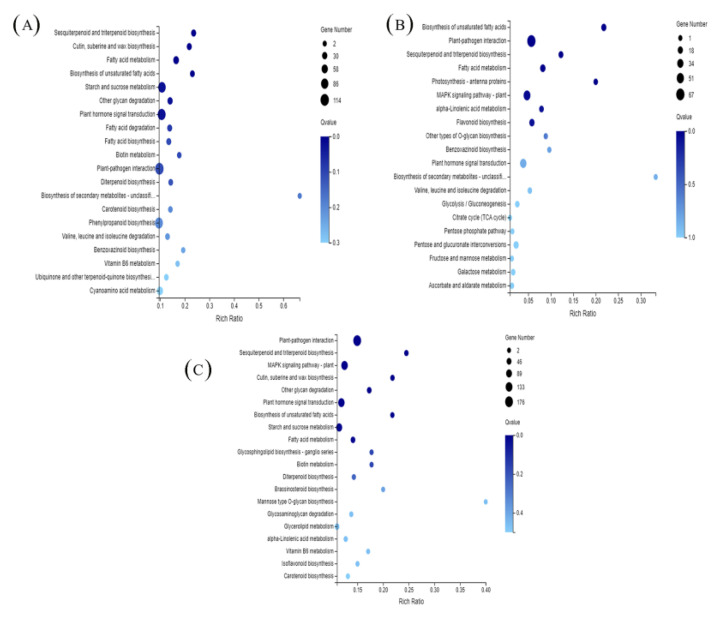
KEGG pathway enrichment analysis of DEGs in response to different stress treatments. (**A**) H vs. CK; (**B**) HS vs. H; (**C**) HS vs. CK. CK: 22 °C/17 °C, distilled water; H: 35 °C/30 °C, distilled water; HS: 35 °C/30 °C, distilled 1 mM Spd.

**Figure 7 antioxidants-11-02332-f007:**
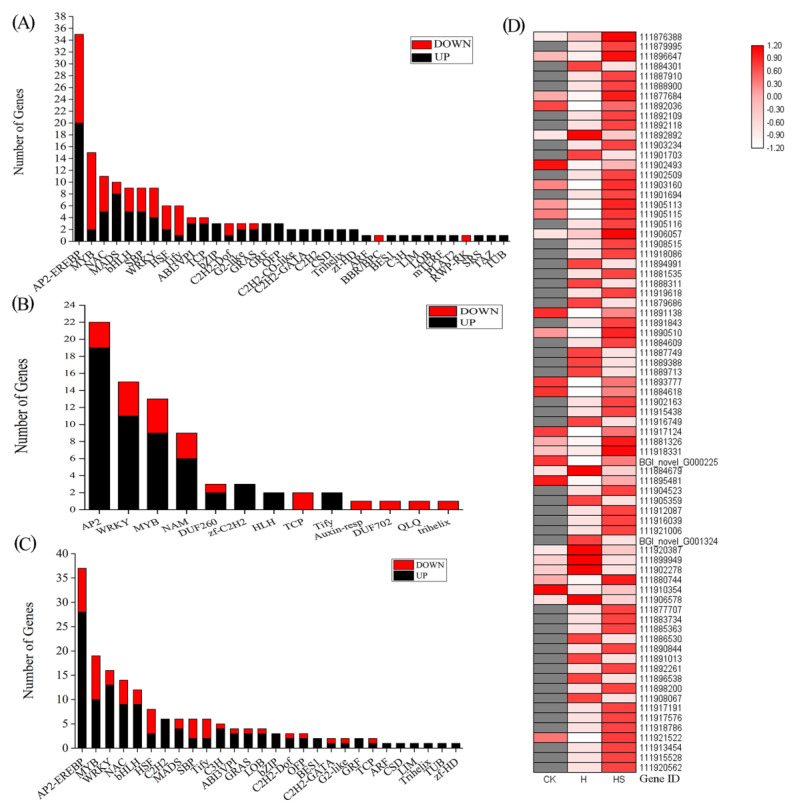
Regulation of DEGs of transcription factors. (**A**) H vs. CK; (**B**) HS vs. H; (**C**) HS vs. CK; (**D**) Expression trends of transcription factors in HS vs. H. CK: 22 °C/17 °C, distilled water; H: 35 °C/30 °C, distilled water; HS: 35 °C/30 °C, distilled 1 mM Spd.

**Figure 8 antioxidants-11-02332-f008:**
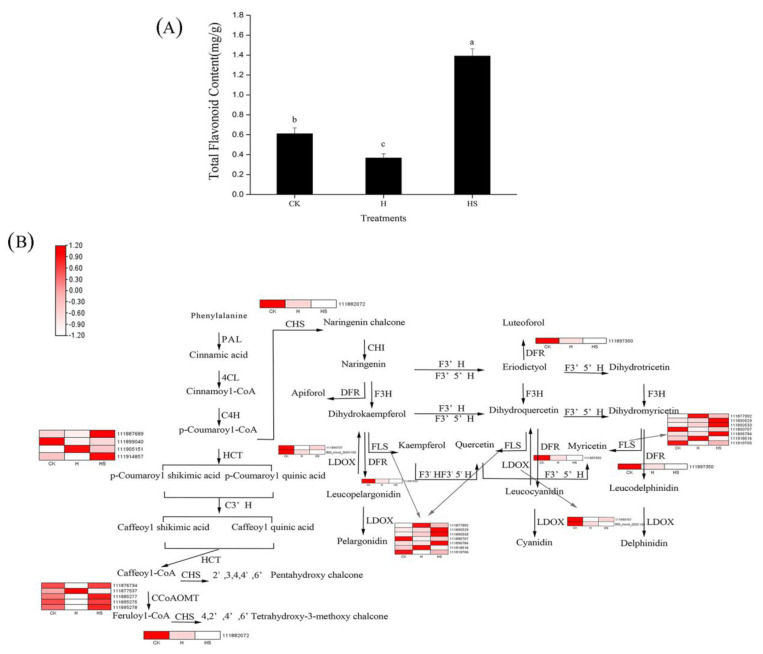
Flavonoid metabolic pathway association with differentially expressed genes (DEGs). (**A**) Total flavonoid content; (**B**) Expression of related DEGs in flavonoid metabolic pathways. Values above each vertical bar followed by different letters show significant differences (*p* < 0.05). The highest value was labeled as a, and those with significant differences were labeled as b, c in that order. CK: 22 °C/17 °C, distilled water; H: 35 °C/30 °C, distilled water; HS: 35 °C/30 °C, distilled 1 mM Spd.

**Figure 9 antioxidants-11-02332-f009:**
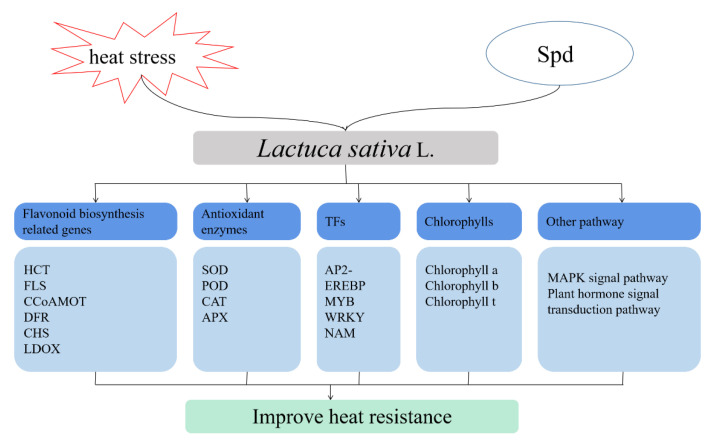
Hypothetical model of exogenous Spd to improve the heat tolerance of lettuce.

**Table 1 antioxidants-11-02332-t001:** Experimental treatments.

Treatment	Temperature (Day/Night)	Type of Spraying
CK	22 °C/17 °C	distilled water
H	35 °C/30 °C	distilled water
HS	35 °C/30 °C	1 mmol·L^−1^ Spd

**Table 2 antioxidants-11-02332-t002:** Effects of exogenous spermidine on lettuce growth under high-temperature stress. Values above each vertical bar followed by different letters show significant differences (*p* < 0.05). The highest value was labeled as a, and those with significant differences were labeled as b, c in that order.

Treatments	Fresh Weight (g)	Shoot Fresh Weight (g)	Root Fresh Weight (g)	Root Length (cm)	Dry Weight (g)	Plant Height (cm)	Root-Shoot Ratio	Dry Weight (g)
CK	38.60 ± 1.46 ^a^	22.78 ± 2.90 ^a^	12.28 ± 0.68 ^a^	25.83 ± 2.25 ^a^	1.01 ± 0.23 ^a^	15.33 ± 2.52 ^a^	1.71 ± 0.25 ^a^	1.01 ± 0.23 ^a^
H	19.01 ± 0.93 ^c^	7.66 ± 0.91 ^b^	11.04 ± 1.26 ^a^	9.61 ± 1.57 ^c^	0.32 ± 0.02 ^c^	16.57 ± 0.40 ^a^	0.58 ± 0.10 ^b^	0.32 ± 0.02 ^c^
HS	24.32 ± 1.54 ^b^	10.98 ± 0.85 ^b^	11.22 ± 0.24 ^a^	14.14 ± 1.36 ^b^	0.61 ± 0.04 ^b^	17.50 ± 1.00 ^a^	0.81 ± 0.03 ^b^	0.61 ± 0.04 ^b^

CK: 22 °C/17 °C, distilled water; H: 35 still, distilled water; HS: 35 °C/30 °C, distilled 1 mM Spd.

**Table 3 antioxidants-11-02332-t003:** PLS-DA model parameters.

Group	R2	Q2
H vs. CK	0.992	0.735
HS vs. H	0.952	0.105
HS vs. CK	0.997	0.811

CK: 22 °C/17 °C, distilled water; H: 35 °C/30 °C, distilled water; HS: 35 °C/30 °C, distilled 1 mM Spd.

## Data Availability

All of the data is contained within the article.

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
