# Peer review of "Transcriptome and Metabolome Analysis Revealed That Exogenous Spermidine-Modulated Flavone Enhances the Heat Tolerance of Lettuce"

_antioxidants, 2022, doi:10.3390/antiox11122332_

Round 1

Reviewer 1 Report

The authors investigate how spermidine confers tolerance to high temperature using the gene network and metabolites analysis in Lettuce.

Thanks to this approach, the authors speculate that this tolerance might come from flavonoids through its antioxidant property. Shading light on a possible mechanistic action of spermidine on plant heat tolerance.

While the approach is quite appropriated, the conclusion are too sharp since the authors are only describing their results but did not directly set up an experiment to tackle their hypothesis. 

Moreover, the authors claims are not supported by the data.

The reference to the figure in the manuscript are not done properly making really hard to follow the author claim. 

The author should pay much more attention before submitting a paper, because otherwise it's not easy at all to do the reviewer job.

Please find the rest of my comments in the document attached.

Reviewer 2 Report

The manuscript deals with effect of spermidine spraying on the flavonoids content by using biochemical traits, enzymes, transcriptome and matabolome analyses under temperature stress.

The topic is of great interest and meets bot the climate current situation and the expectation of th Antioxidants.

However, that are many shortcomings in this manuscript.

1- As other studies based on metabolome and transcriptome, the work is based on one experiment, which is very limiting factor. Indeed, the timing and the intensity of temperature stress may influence the response of plants. This point is neither evoked nor discussed in this manuscript

2- lack of literature background induces the gaps in the choice of measured biochemical traits. Indeed it is well known that ALL photosynthetic pigments are involved in the plant response to stresses. This is particularly true for carotenoids ans xantophylls  under temperature stress. Unfortunately, authors did not measure these pigments. Moreover, the authors erroneously use the term "photosynthetic pigments" in their discussion and conclusion. In fact, this is wrong since they only measured chlorophylls.

3-the developed discussion is rather correlative. It is expected a mechnistic. when read the abstract, a course of action is proposed. But when read in full, nothing is presented about the mode of action of spermidine on flavonoid biosynthesis and even less on chlorophylls (not photosynthetic pigments).

Please replace photosynthetic pigments by chlorophyll (more precise) in the whole manuscript.

Further remarks:

Abstract

L25-27. Please rephrase this sentence. Authors did not elucidate the mechanism of action but only have shown correlations and observed DEGs regulation. Thery did not propose a mechnistic explanation.

L91-92. « The heat-sensitive lettuce variety 'Beisansheng No. 3' was selected from Beijing University of Agriculture. ». Please add a reference

L98-101. This part is not clear. Please display the dose of spermidine received by lettuce seedlings.

The concentration of the solution to be sprayed is at 1 mmol·L-1. How many mL were sprayed on each lettuce seedling ?

This is important since the effect of spermindine (as plant growth regulator) is dore dependant.

L102-106. Please transform this paragraph into a table. It will be clearer.

Authors have measured chlorophylls contents.

Why didn't the authors determine the carotenoid content? This is an important traits of plant reaction to high temperatures.

Please delete L209-211.

Table 1 means comparison test are wrong for  root length and for Root-shoot ratio .

Indeed, if CK presented the highest value for root length and the letter a has been attributed to this mean value, Treatment H prensented the lowest value and authors have attributed the letter b which is not right. Indeed the treatment HS, with a mean of 14.14 is an intermedate value, have received the letter c and can therefore considered lower than 9.61 (treatment H). This is without any sense.

Please revise all the table.

 This is also the case for APX activity in table 2. Please revise

Round 2

Reviewer 1 Report

I would like to thank the authors for addressing all the comments.

Congrats!

Just noticed a typo: In the table 2 legend, typo: “H35still, distilled water;”

Reviewer 2 Report

I am satisfied with the changes made in the manuscript and the consideration of the remarks I made